# Time-Domain Joint Training Strategies of Speech Enhancement and Intent Classification Neural Models

**DOI:** 10.3390/s22010374

**Published:** 2022-01-04

**Authors:** Mohamed Nabih Ali, Daniele Falavigna, Alessio Brutti

**Affiliations:** 1Information Engineering and Computer Science School, University of Trento, 38121 Trento, Italy; 2Fondazione Bruno Kessler, 38121 Trento, Italy; falavi@fbk.eu (D.F.); brutti@fbk.eu (A.B.)

**Keywords:** joint training, speech enhancement, intent classification

## Abstract

Robustness against background noise and reverberation is essential for many real-world speech-based applications. One way to achieve this robustness is to employ a speech enhancement front-end that, independently of the back-end, removes the environmental perturbations from the target speech signal. However, although the enhancement front-end typically increases the speech quality from an intelligibility perspective, it tends to introduce distortions which deteriorate the performance of subsequent processing modules. In this paper, we investigate strategies for jointly training neural models for both speech enhancement and the back-end, which optimize a combined loss function. In this way, the enhancement front-end is guided by the back-end to provide more effective enhancement. Differently from typical state-of-the-art approaches employing on spectral features or neural embeddings, we operate in the time domain, processing raw waveforms in both components. As application scenario we consider intent classification in noisy environments. In particular, the front-end speech enhancement module is based on Wave-U-Net while the intent classifier is implemented as a temporal convolutional network. Exhaustive experiments are reported on versions of the Fluent Speech Commands corpus contaminated with noises from the Microsoft Scalable Noisy Speech Dataset, shedding light and providing insight about the most promising training approaches.

## 1. Introduction

Recently, the use of audio-visual platforms e.g., Microsoft Teams, Google Meet, Zoom, etc., for smart-working, remote collaborations and many other applications has been growing exponentially. In these cases, the speech signal is the predominant tool used for communication, and sharing ideas between people [1]. Unfortunately, in these application scenarios, speech signals are usually corrupted by environmental noises or by the presence of other sound sources, e.g., TV, or other speakers in cocktail party scenarios [2,3]. Despite the presence of this interference, humans have the ability to extract the target speech signals, while ignoring noises and other interfering signals [4,5]. Unlike humans, many speech applications, like Automatic Speech Recognition (ASR), suffer in the presence of these adverse noisy conditions which deteriorate the speech quality and intelligibility, leading to considerable performance drops [6,7], especially in low level of signal-to-noise ratio (SNR).

In recent years, substantial progress has been made to mitigate the noise effect. A possible approach is to train, or adapt the models on the noisy data [8]. This can be done either by collecting application specific data or through the usage of data augmentation strategies [9]. However, it has to be considered that gathering large noisy datasets is costly and time consuming while, in general, all possible noisy conditions cannot be known a-priori making unfeasible the data augmentation based approach. Therefore, an alternative method is to use a speech enhancement (SE) front-end to improve the speech quality and intelligibility. SE, which is implemented as computer algorithms able to extract the target speech from noisy mixtures, is a fundamental task in the field of speech processing, and is currently integrated in wide range of applications such as: mobile telecommunication, speaker recognition and ASR systems. During the last decades, tremendous growth has been observed in the speech enhancement research area, in particular towards improving the robustness of ASR systems in noisy conditions. Figure 1 shows a general diagram of a speech enhancement system.

Mathematically, denoting x[n] as the clean speech signal and s[n] as the additive noise (the environmental noise is also hypothesized to be additive) at time index *n*, then the noisy speech signals y[n] can be expressed as:(1)y[n]=x[n]+s[n]

The goal of speech enhancement algorithms is to estimate the enhanced signal $\hat{x}\left[n\right]$ from the noisy signal y[n], such that:(2)$$\hat{x}\left[n\right]\approx x\left[n\right]$$

Classic techniques based on signal processing are, for example, spectral subtraction, Wiener filter for single channel scenario and beamformers, which employ microphone arrays to further reduce the noise effect [10]. In particular, the minimum variance distortion-less response (MVDR) beamformer [11] is the most common solution, that employs a multi-channel Wiener filter [12] as post-filter. Unfortunately, the effectiveness of these techniques in reducing the impact of noise, i.e., improving the signal-to-distortion ratio (SDR) and the SNR, often does not lead to an improvement of ASR accuracy in terms of the word error rate (WER). In addition, signal processing methods perform poorly in presence of highly non-stationary noise, mainly because they rely on estimating of static spectral properties of the noise component [13].

Recently, the progress of deep learning algorithms has brought substantial improvements also in the SE field [14,15,16,17,18,19]. Deep learning techniques are data-driven approaches that frames the SE task as a supervised learning problem aiming at reconstructing the target speech signals from the noisy mixture. A very popular set of neural spectral-based methods employ neural networks to estimate Time-Frequency (T-F) masks which are then used to separate the T-F bins associated to the target source and the noise. The network is trained using either ideal binary mask (IBM) or ideal ratio mask (IRM) as training targets [20,21]. Typically, the networks are trained using the mean squared error (MSE) either on the masks or on the reconstructed signal [22,23]. Despite the promising performance achievable in terms of SDR and intelligibility, the presence of artifacts in the reconstructed signals compromises the performance of further processing stages. In addition, only the magnitude of the spectrogram is enhanced, while the phase is left unprocessed.

One solution that can mitigate these issues is to implement the enhancement task in the time domain and process the raw waveforms [24]. In this work, we aim at investigating possible ways to optimize the front-end speech enhancement not only in terms of signal quality but also to take into account the performance of the subsequent back-end component.

In particular, we address the intent classification task on noisy data and we propose a pipeline that integrates Wave-U-Net [25], a time-domain enhancement approach, with an end-to-end intent classifier implemented with a time-convoluted neural model.

Our contribution to this task is to investigate methods to jointly optimize the front-end speech enhancement for noise removal and the back-end intent classification task. To the best of our knowledge, this the first attempt to jointly train an end-to-end neural model for both speech enhancement and intent classification in the time domain. This paper extends our previous preliminary work published in [26], where we investigated the use of pre-trained speech enhancement models in combinations with intent classification. With respect to this work, the previous paper does not consider joint training of the two model. Moreover, the back-end operates in the frequency domain.

The rest of the paper is organized as follows: Section 2, we survey the recent joint training approaches for different speech tasks. Section 3, we explain each component in the proposed system description. In Section 4, we report and discuss our experimental results. Finally, in Section 5 we conclude our work.

## 2. Related Work on Jointly Training of Speech Enhancement with Different Speech Tasks

Three main strategies can be considered to incorporate a speech enhancement front-end into different speech-based applications. The first one consists in training the back-end based component (i.e., the IC classifier in the case of this work) on clean speech signals, while at inference stage a speech enhancement front-end is integrated for noise removal [27]. The main disadvantage of this approach is represented by distortions introduced by the front-end (i.e., the speech enhancement module) that didn’t occur in the training data of the back-end. However, this strategy is still beneficial in different noise-robust speech based applications.

To overcome this limitation, the second strategy filters the training data of the back-end with speech enhancement, so that the back-end component works on the enhanced features. This strategy is useful in strengthen the back-end against noise, but it is highly dependent on the speech enhancement performance [28]. In general, it was found to be better training the back-end on noisy data-sets, if they contain enough samples of the noise present in the operating conditions on the field.

The third strategy is to make the back-end component work on the noisy speech, while at inference noisy features are fed either to the back-end directly or, first, to the speech enhancement module. These multi-condition training strategy has shown promising results in [29], but its performance in unmatched conditions is poor [30]. Each strategy has its own advantages and disadvantages, and it is highly dependent on the application domain.

Jointly training, the strategy proposed in this paper aims to jointly adjusting the parameters of a neural (front-end) speech enhancement model and a neural (back-end) model designed for a specific task (e.g., ASR, voice activity detection, or intent classification). Thus, the speech enhancement front-end provides the “enhanced speech” desired by the back-end model. In this way the back-end model guides the front-end towards the execution of a better discriminative enhancement. Figure 2 shows the schematic diagram of a conventional joint training approach including speech enhancement and an end-to-end (E2E) back-end. Note in the figure the two different losses defined for the front-end and the back-end that will be combined in a global loss as will be explained in Section 3.3.

In the next sub-sections, we survey the most recent research based on joint training speech enhancement with other back-ends e.g., ASR and voice activity detection.

### 2.1. Jointly Training Speech Enhancement with Voice Activity Detection (VAD)

Many research works had been conducted to improve the VAD performance in low SNR environments. Typically the basic solution is to integrate a SE front-end as a pre-processing stage to eliminate noise [31].

The authors of [32,33] trained the VAD based on the the denoised speech signals obtained from the SE front-end in which both the front-end model and the VAD model are jointly optimized and fine-tuned. Later it was observed that, training VAD directly based on the denoised signals resulting from the SE front-end may decrease the VAD performance, especially when the performance of the SE is poor [34]. To mitigate this effect several researches integrates advanced SE front-end to extract the denoised features for VAD training [35].

Inspired by the performance of U-Net in the field of medical imaging segmentation [36], the authors in [37] integrated a SE front-end based on U-Net to estimate both clean and noise spectra simultaneously, while the the VAD is trained directly on the enhanced signals.

Another contribution is done in [38], in which a variational auto-encoder (VAE) is used to denoise the speech signals while the VAD is trained on the latent representation of the VAE. The authors of [34], instead of training the VAD on the latent representation of the VAE, concatenate the noisy acoustic features with the enhanced features estimated from a convolutional recurrent neural network.

Finally, Refs. [39,40] proposes the use of a multi-objective network to jointly train SE and VAD to boost their performance. In this system both modules share the same network with different loss functions. Unfortunately, this technique weaken the performance of the VAD module.

### 2.2. Jointly Training Speech Enhancement with ASR

An early attempt for jointly training speech enhancement with ASR was proposed in [41], where a front-end based feature extraction was jointly trained with a Gaussian Hidden Markov Model back-end and optimized with a maximum mutual information criterion. Towards this direction, the authors in [42] proposed a novel jointly training approach by concatenating a deep neural network (DNN) based speech separation module, a feature extractor based on a filter-bank and an acoustic model, and train these models jointly.

The authors of [43], beside investigating feature mapping based on DNN, jointly trained a single DNN for both feature mapping and acoustic modelling. The proposed approach showed a clear improvement in the ASR performance.

In [44] the authors addressed the problem of joint training when the front-end output distribution change dramatically during model optimization, noticing a performance drawback on the ASR task due to the fact that the back-end needs to deal with a non-stationary input. To mitigate this effect, the authors proposed a a joint-training approach based on a fully batch-normalized architecture.

Inspired by its performance in the computer vision, the authors of [45] investigated the usage of generative adversarial networks (GAN) in the speech enhancement area. They proposed a joint training framework based on adversarial training with self-attention mechanism for ASR noise robustness. The proposed system consists of a self-attention GAN for speech enhancement with a self-attention end-to-end ASR model. Finally, a recent research in [46] proposes a joint training approach based on a gated recurrent fusion (GRF) for ASR noise robustness.

## 3. System Description

As previously mentioned, this paper tackles the problem of intent classification (IC) in noisy environments considering the combination of a speech enhancement component, based on the Wave-U-Net architecture [25], and a time convolutional network (TCN) [47] that performs the intent classification task. Figure 3 shows the complete pipeline of the proposed system. The following subsections describe each module in details.

### 3.1. Wave-U-Net for Speech Enhancement

Recently, time-domain approaches for speech enhancement operating directly on the raw wave-forms have been proposed [48,49,50,51]. Among them, U-Net [52] was successively improved towards Wave-U-Net [25], allowing to achieve promising results in comparison with other approaches.

The Wave-U-Net model consists of 3 components [53]: (a) an encoder made by multiple 1-D fully convolutional down-sampling blocks; (b) a 1-D convolutional layer called bottleneck layer; and (c) a decoder made by a series of 1-D fully convolutional up-sampling blocks (this architecture is depicted in Figure 3). Note that in this architecture skip connections (i.e., the green arrows in the left part of Figure 3) are applied between each down-sampling layer and the corresponding up-sampling layer of the model.

In details, the input to Wave-U-Net is a noisy signals y[n], n=0,…,L−1, where *L* is the number of samples. During training, low-dimensional high level features are computed at different time scales using a series of downsampling blocks. These features are then concatenated with its corresponding local, and high resolution features computed from the upsampling blocks. In case of monaural speech enhancement, the network is trained to map the noisy signal y[n] to its enhanced counterpart $\hat{x}\left[n\right]$ using the clean signal x[n] as the training targets, by minimizing the mean squared error (MSE) loss between x[n] and $\hat{x}\left[n\right]$, i.e.,:(3)$$L_{SE}=\sum_{n}{∥x\left[n\right]-\hat{x}\left[n\right]∥}^{2}$$

### 3.2. Intent Classification

The IC task aims to recognize the intents encoded in a given spoken utterance [54]. This task is usually implemented by processing the outputs of an ASR system with natural language processing (NLP) tools, in order to produce a semantic interpretation of the input speech. For example, in smart home applications an utterance like “increase the sound” might correspond to an intent represented with the following filled slots: action: “increase”, type: “sound”, count: “None”, place: “None”. A survey reporting fundamentals of spoken language understanding (SLU) technology can be found in [55,56].

Recently, different approaches that perform this task in an E2E fashion have been investigated with excellent performance on several data sets. The E2E paradigm uses a single neural model to map a spoken input into the corresponding intents, thus optimizing directly the classification metrics and avoiding error propagation caused by ASR errors. Some interesting models and related results in this direction can be found in the works reported in [57,58,59,60]. These approaches have been proved to be effective both on large data sets, such as Google Home [58,61], and on a smaller data set, such as the Fluent Speech Command [57]. As reported in [57], the reason for this is manifold: (a) E2E models avoid using either useless information or information contaminated by errors in the ASR output; (b) it learns directly the metric used in the evaluation phase; and (c) it can take advantage from supra-segmental information contained in the speech signal to process.

In this work, we use the model described in Figure 4a, which is based on Conv-TAS Net, a neural architecture introduced for time audio separation [47]. The model processes the enhanced signals from the SE front-end as input. The model architecture consists of a normalization layer followed by a 1-D convolutional layer that maps the input features into bottleneck features with 64 channels. The input layer is followed by 2 convolutional blocks, each of them includes 5 residual blocks, with SoftMax activation function. Each residual block consists of 1-D dilated convolutional layers, normalization layers with Parametric Rectified Linear Unit (pReLU) activation function as shown in Figure 4b. The dilation factor is increased for each successive residual block. Then, 64 channels for bottleneck features and 128 channels for depth-wise separable convolutional layers are used, respectively.

Skip connections are also used between each residual block. The IC classifier is trained to predict the target intent by minimizing the cross entropy loss between the actual and predicted labels $L_{IC}$, i.e.,:(4)$$L_{IC}=-\frac{1}{T}\sum_{t}\log\left(p_{t}\right)$$
where *T* is the number of training samples and $p_{t}$ is the probability of the *t*th target sample.

### 3.3. Joint Training Architectures

Training the two components in a joint way typically provides better performance than training the models independently [38,40]. Therefore in this work we investigate different model architectures varying the interconnection between the speech enhancement and the intent classifier components, as shown in Figure 5. The **Joint Training** (JT) approach, depicted in Figure 5a, is the most straightforward combination strategy where the IC module receives as input the enhanced signals. The **Bottleneck** approach (BN) is depicted in Figure 5b. Instead of the reconstructed time domain signal, this combination uses the bottleneck features of the SE component as input to the intent classifier. Finally, Figure 5c shows the **Bottleneck-Mix** (BN-Mix) strategy: a more articulated combination approach which concatenates the noisy mixture with the bottleneck representations.

All the three end-to-end joint approaches depicted in Figure 5 were trained using a compound loss defined as follows:(5)$$L=\alpha L_{SE}+\left(1-\alpha \right)L_{IC},$$
where $L_{SE}$ and $L_{IC}$ are the MSE loss and the cross-entropy loss for speech enhancement and intent classification, respectively. The coefficient α∈(0,1) is a hyper-parameter that adjusts the weight of each component in the joint loss. In all proposed strategies we measure the performance on a grid of values for α, i.e., (0, 0.1, 0.5, 0.9).

Although the loss is the same, its components (i.e., $L_{SE}$ and $L_{IC}$) affect differently the model parameters depending on the particular architecture and providing different performance trends for both *SE* and *IC* tasks, as will be shown in Section 4.2. As a matter of fact, the gradient of the *IC* loss acts in different manners across the the parts of the SE model. These (*SE*) model parameters $\Theta_{SE}$ are updated as follows:(6)$$\Theta_{SE}\overset{}{\leftarrow }\Theta_{SE}-\lambda_{1}\left[\alpha \nabla L_{SE}+\left(1-\alpha \right)\nabla L_{IC}\right],$$
where $\nabla L_{SE}$ and $\nabla L_{IC}$ represent the gradients of *SE* and *IC* respectively, and $\lambda_{1}$ is the learning rate for the SE front-end. As a consequence, the front-end is expected to produce enhanced signals not only for matching the target clean signals but also to maximize the *IC* performance. Note that in BN and BN-mix training strategies the decoder part of the SE module is not affected by the $L_{IC}$. Unlike the front-end, the IC module depends only on its own loss function and its parameters are updated as:(7)$$\Theta_{IC}\overset{}{\leftarrow }\Theta_{IC}-\left[\left(1-\alpha \right)\lambda_{2}\nabla L_{IC}\right],$$
where $\Theta_{IC}$ denotes the the *IC* parameters, and $\lambda_{2}$ is the *IC* learning rate.

## 4. Experimental Analysis

### 4.1. Dataset

For our experimental analysis we consider the Fluent Speech Commands (FSC) dataset [57]. FSC includes 30,043 English utterances, recorded as 16 kHz single channel .wav audio files. Overall, 97 native and non native speakers (52 males, and 49 females) are recorded while simulating an interaction with voice-enabled appliances or smart devices with no overlapping speech in each signal. Overall, the dataset provides 248 different utterances that are mapped in 31 different intents. Each intent consists of three items: action, object, and location. For example, “increase heat in the kitchen” is categorized as: action: “increase”, object: “heat”, location: “kitchen”. Totally, 6 different actions, 14 objects and 4 locations are available. For training, validation and evaluation, we consider the official splits as described in Table 1 [57]. In order to avoid the presence of long silence in the wav files, all signals were cut using librosa.effects.trim() module.

To emulate realistic application scenarios, where environmental noise may affect the intent classification accuracy, we contaminated the FSC dataset creating an equivalent noisy version. In particular, we superimpose 6 different types of noise out of the 25 noises available in the Microsoft Scalable Noisy Speech Dataset (MS-SNSD) [62]: e.g., air conditioner, airport announcements, traffic, neighbor speaking, shutting door, and restaurant.

The FSC dataset is contaminated using the “maracas” library available in [63]. For each clean utterance, a noise signals is randomly selected from those available and superimposed with an SNR randomly selected from 3 possible values: −5 dB, 0 dB, 5 dB. Note that the resulting noisy dataset includes a uniformly distributed variety of different conditions in terms of type and amount of noise.

The speech enhancement network is designed to process fixed-length input signals with sample length 16,834 [64]. Therefore, signals longer than that are first split in segments and then concatenated at the output of Wave-U-Net. The model has 12 layers. The encoder part uses with kernelsize=15, stride=1, and padding=7, while the decoder has the same number of layers but with kernelsize=5, stride=1, and padding=2. Each layer in both encoder and decoder is followed by a 1D batch normalization layer, and leaky ReLU activation function with negative slope equals to 0.1. The model is trained by optimizing the MSE loss with learning rate $\lambda_{1}=10^{-4}$. For the intent classifier, the model is trained by optimizing the cross entropy loss with learning rate $\lambda_{2}=10^{-3}$. For the BN-MIX experiment the first layer is 1-D convolutional layer with kernelsize=1. Both models are trained with ADAM optimizer with decay rates are β1=0.9 and β2=0.999, batch size 2.

Although our final goal is to improve the classification accuracy, we also evaluate the performance of the enhancement component. We consider traditional speech quality metrics, namely PESQ [65], STOI [66], and MSE. The PESQ metric is based on the wide-band version recommended in ITU-T [67], and its range is extended from −0.5 to 4.5. STOI is based on a correlation coefficient between time-aligned clean and enhanced signals, and its range is from 0 to 1. For both metrics the highest score is the best. We also consider the MSE metric as a similarity measurement between the clean and enhanced signal. Unlike the PESQ and STOI, a lower MSE is better. Finally, the performance of the back-end is evaluated using the intent classification accuracy, that measures the actual match between the estimated intent slots and it corresponding ground truth ones.

### 4.2. Experimental Results

Table 2 reports the classification accuracy resulting from the different joint-training approaches described above, using different values of the parameter α in Equation (Equation 5). The table reports also the results of JT applied to clean signals as upper bound. The column “noisy” refers to the performance obtained providing the noisy signals to the back-end. From the table it appears evident the improvement brought by the JT model. The highest classification accuracy is achieved with α=0.5, indicating that SE loss and IC loss equally contributes. Note that α=0 correspond to training the model without SE, i.e., the SE front-end is part of the classifier. This improves the classification accuracy with respect to the noisy case as the classifier is actually deeper. When α is very large (i.e., 0.9) the front-end tends to give more importance to the signal reconstruction rather than making the output signal suitable for the IC component. Thus the final classification performance drops. Interestingly, the $L_{SE}$ loss helps improving the performance also when JT is applied to clean signals. In the latter case, when α=0 only a small improvement is observed with respect to the noisy case, while for larger values the model reaches almost state of the art performance (achieved with larger models or using spectral features).

A similar trend is not observed in both the other two architectures: BN and BN-mix. The BN approach seems to be not influenced by α and performs very similarly to JT with α=0. The fact that the encoder of Wave-U-Net does not interact with the IC component make the contribution of $L_{SE}$ negligible. In addition, the bottleneck representation is probably too compact and focused on other properties of the signal. For the BN-mix method, since the difference in dimensionality between the combined features (bottleneck and 1D-Conv output) is very high the contribution of the bottleneck is very limited. In facts, the performance is just slightly better than the noisy baselines. Note that for very high values of α the front-end starts to provide some beneficial influence. Finally, we experiment on alpha=1, but these experiments do not bring any improvement in the accuracy as the back-end is not trained.

Table 3, Table 4 and Table 5 show the SE performance achieved with the various architectures. As previously observed, both BN and BN-Mix strategies optimize the Wave-U-Net decoder independently on the $L_{SE}$ loss. Hence, it is of no surprise that the resulting intelligibility metrics are much higher than JT. Note however that the enhancement is not dependent on α as long as it is larger than 0. Basically the decoder is capable of reconstructing the signal counterbalancing the impact of the IC on the encoder. For what concerns JT, as α increases the signal reconstruction increases. Finally, it is remarkable to observe that reconstruction quality and classification accuracy are in contrast with each other and it is not possible to effectively optimize both.

In summary, looking at the performance reported in Table 2 we notice that, as expected, the best IC performance is achieved with the JT approach, which carries out optimization directly on the enhanced signal at the cost, however, of reduced enhancement results. On the contrary both the BN and BN-mix, working on the bottleneck of the Wave-U-net model, gives better enhancement quality at the cost of lower IC accuracy. Finally, for better interpretation, we report all the evaluation metrics in graphical representation as shown in Figure 6. Also, we report all the abbreviations in Table A1, see Appendix A.

## 5. Conclusions

In this paper we proposed an end-to-end joint training approaches to robust intent classification in noisy environment. The jointly compositional scheme consists of a neural speech enhancement front-end based on Wave-U-Net combined with an end-to-end intent classification scheme. In particular, we investigate three different joint training strategies which combine the two components in different ways, namely JT, BN, and BN-Mix.

All experiments are conducted on the FSC dataset contaminated with a set of noises from MS-SNSD. Contrary to what observed in other speech related classification tasks, experimental results validate the efficacy of the proposed joint training approach, in which de-noising actually is beneficial in terms of final classification accuracy when models are trained on matched noisy material. We observed that equally balancing the enhancement and classification losses gives the best results. In addition, it is worth noting that injecting an intermediate loss is always beneficial, also with clean data. The motivation could be that given the large size of the model and the relatively small amount of training material the intermediate loss guides the network towards its optimal configuration. Finally, we also observed that the sequential nature of JT is better than the multi-task structure used in BN and BN-mix.

### Future Directions

One future direction is to evaluate the proposed approach on different more complex datasets, as for example the ATIS corpus [68], the Almawave-SLU corpus [69], the SLURP corpus [70]. In addition, to better assess the robustness and flexibility of the proposed approach, we plan to apply the same joint training scheme to other speech processing tasks, eventually involving seq2seq or regression tasks. One option is also to include multiple parallel tasks and jointly train the model in a multi-task learning fashion. Finally, we plan to investigate the joint training approach in combination with speech embedding. In particular, wav2vec pre-trained models [71] can be used as an intermediate stage between the front-end and back-end models i.e., to extract the speech embedding form the enhanced speech signals and train the back-end based on these embedding. An alternative approach, is to integrate the wav2vec model on top of the speech enahncement front-end i.e., the front-end estimates the enhanced speech embedding, that will be later used to train the back-end.

## Figures and Tables

**Figure 1 sensors-22-00374-f001:**
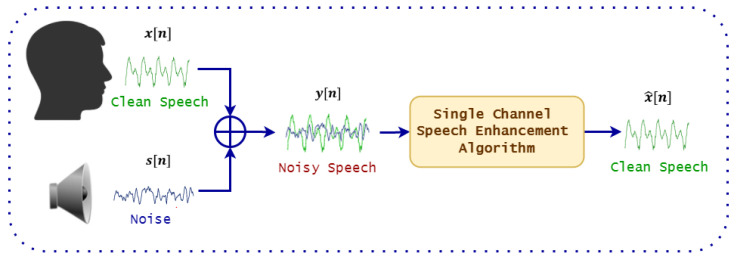
The basic diagram of speech enhancement system.

**Figure 2 sensors-22-00374-f002:**
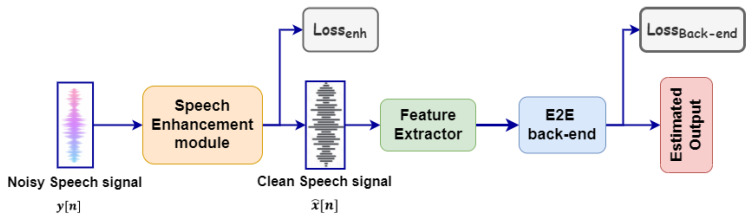
Schematic diagram of the conventional joint training speech enhancement with different back-end.

**Figure 3 sensors-22-00374-f003:**
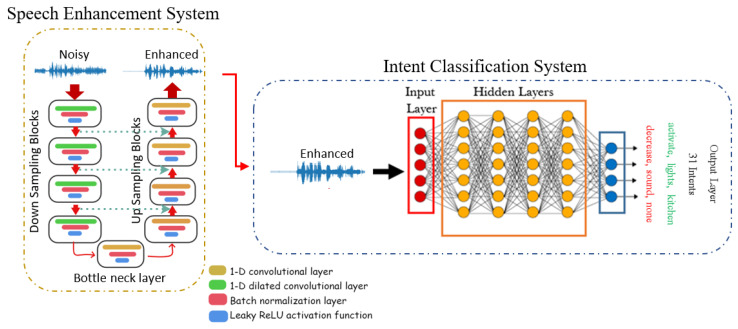
The full pipeline of our intent classification scheme, including speech enhancement and intent classifier.

**Figure 4 sensors-22-00374-f004:**
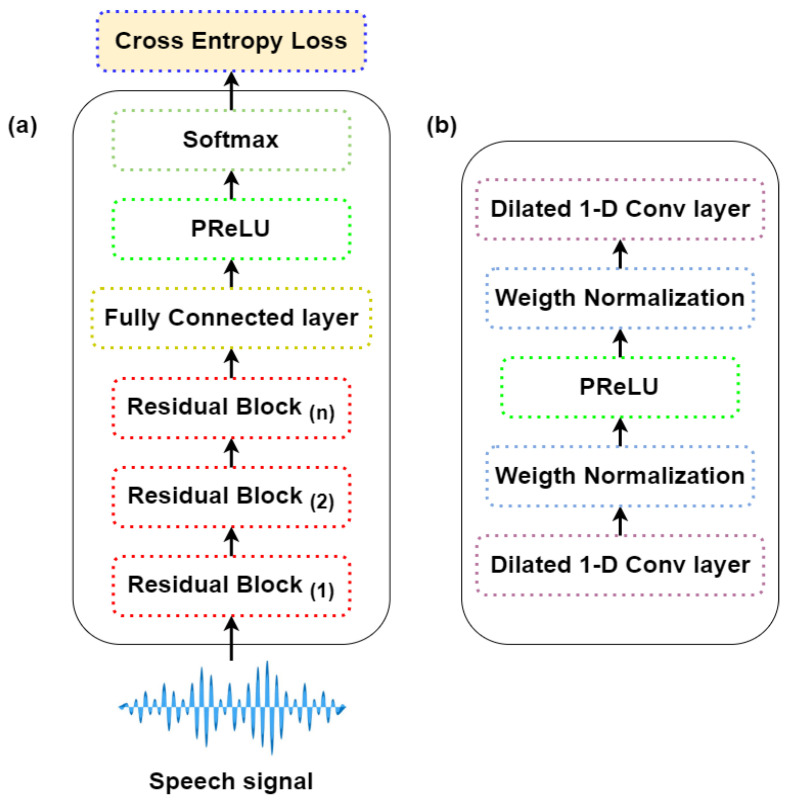
The general architecture of: (**a**) Time Convolutional network for intent classification. (**b**) The architecture of the residual block.

**Figure 5 sensors-22-00374-f005:**
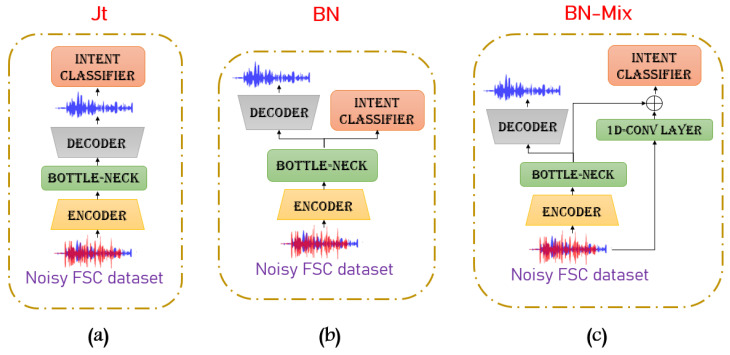
Three strategies of the proposed joint training approaches: (**a**) based on the mixture signals (JT). (**b**) based on bottleneck representation (BN). (**c**) based on the concatenation between mixture signals and bottleneck representation (BN-Mix).

**Figure 6 sensors-22-00374-f006:**
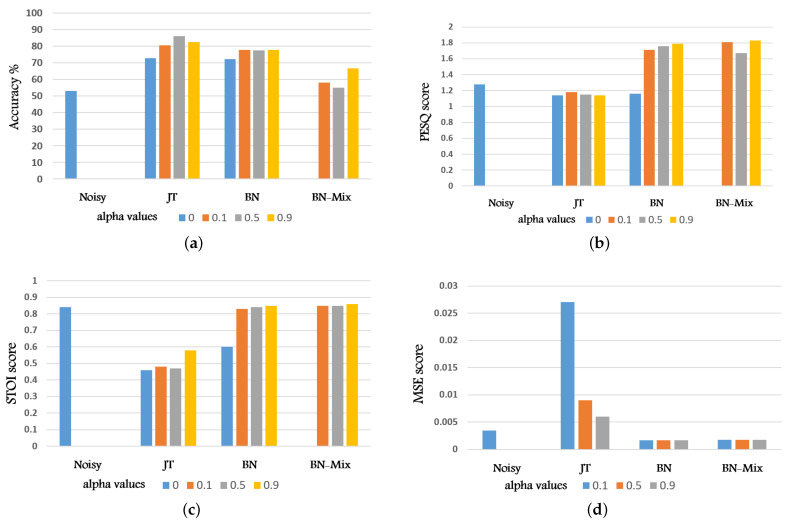
Graphical representation for: (**a**) the classification accuracy. (**b**) PESQ metric. (**c**) STOI metric. (**d**) MSE metric against α values for all experiments.

**Table 1 sensors-22-00374-t001:** FSC dataset Description.

Data	Speakers No.	Utterance No.	Total Hours
Train Data	77	23,132	14.7
Validation Data	10	3119	1.9
Evaluation Data	10	3793	2.4

**Table 2 sensors-22-00374-t002:** Classification accuracy for different architectures with different α.

Noisy		Jt-Clean	Jt	BN	BN-Mix
53.2%	α=0	73.37%	72.80%	72.39%	-
	α=0.1	91.53%	80.50%	77.80%	58.02%
	α=0.5	92.77%	86.02%	77.53%	54.99%
	α=0.9	-	82.52%	77.90%	66.67%

**Table 3 sensors-22-00374-t003:** The PESQ metric for different architectures with different α.

Noisy		JT	BN	BN-Mix
1.28	α=0	1.14	1.16	-
	α=0.1	1.18	1.71	1.81
	α=0.5	1.15	1.76	1.67
	α=0.9	1.14	1.79	1.83

**Table 4 sensors-22-00374-t004:** The STOI metric for different architectures with different α.

Noisy		JT	BN	BN-Mix
0.84	α=0	0.46	0.60	-
	α=0.1	0.48	0.83	0.85
	α=0.5	0.47	0.84	0.85
	α=0.9	0.58	0.85	0.86

**Table 5 sensors-22-00374-t005:** The MSE metric for different architectures with different α.

Noisy		JT	BN	BN-Mix
$3.5\times 10^{-3}$	α=0	$7.6\times 10^{-1}$	$1.4\times 10^{-1}$	-
	α=0.1	$2.7\times 10^{-2}$	$1.7\times 10^{-3}$	$1.8\times 10^{-3}$
	α=0.5	$9\times 10^{-3}$	$1.7\times 10^{-3}$	$1.8\times 10^{-3}$
	α=0.9	$6\times 10^{-3}$	$1.7\times 10^{-3}$	$1.8\times 10^{-3}$

## Data Availability

The data that support the findings of this study are available from the corresponding author upon reasonable request.

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
