# Peer review of "Time-Domain Joint Training Strategies of Speech Enhancement and Intent Classification Neural Models"

_sensors, 2022, doi:10.3390/s22010374_

Round 1

Reviewer 1 Report

Remarks:

The paper uses a number of abbreviations. While they are probably known to the specialists, they may however confuse some other readers. The authors should introduce abbreviations only if it is necessary or abbreviation is widely used and known, otherwise, introducing the abbreviation that is used once or twice only is confusing.  

Some of the abbreviations are not introduced at all (NLP, SLU, DNN), some are introduced far after the first use (e.g. E2E). Please check every abbreviation. Please consider whether the list of abbreviations would be helpful.

I personally think that if the signals are named in the text (like x[n], etc.) these names should be also used in figures and diagrams to clearly define where these signals are used. The diagrams could be larger, their quality is too low.

Figure 4 should be referred to at least twice (4a and 4b). The names nReLU and Softmax should be explained either in the figure caption or in the text. The description in lines 177-183 is unclear and should be illustrated with the diagram (or it is in Figure 4?)

Generally the language of the paper seems to be correct, however, the part 3.2 negatively outstand from the rest of the paper, maybe it should be check.

174: E2E models avoid (not avoids) 

 182: double “the”, the verb is missing

(167: have started to be investigated – sounds strange)

other minor remarks:

75: before “ Section 5” „in” is missing

 157,158: should be Figure instead of figure 

next to 184: Figure 5 instead of figure 5

240: double “of”

Reviewer 2 Report

Overall, this is an interesting paper, focused on speech signal processing. As mentioned by the Authors, recently there has been a huge rise in popularity of audio-visual platforms. They aid in remote collaboration as well as various other types of activities. More and more companies worldwide offer dedicated tools and complex solutions. And still, the biggest differentiator is the quality of transmitted speech itself. Any interruption, e.g. caused by environmental noise and/or network equipment, processing, etc., deteriorates the subjective judgement of the user, making it harder to extract information.

The text is written in proper English, it is pleasant to read. The first theoretical part is informative, providing an adequate background to the discussed topic. Presented equations and mathematical formulas seem plausible and free of error. The novelty of the paper is clearly highlighted. Displayed figures and block diagrams are sharp and easy to read. Despite the fact that the research part seems properly designed, some important information seem missing. Generally speaking, some modifications could be done in order to raise the quality of the manuscript.

Suggestions and comments:

  • Minor editorial and formatting issues, e.g. lack of space between subsequent words, etc.
  • Additionally, not all lines of text seem to be numbered – e.g. see page 2. (?)
  • In my opinion, Authors should mention more recently published papers on speech processing and audio coding, etc. Consider extending the list of cited references.
  • It would be advisable to provide more info about the utilized set of speech signal samples. What format were they coded in? What was their bitrate, sampling frequency, sampling rate? Where there any modifications applied before and after processing, etc.? Did they include male or female lectors? Or maybe were they composed of a mixed conversation (multiple/simultaneous lectors)?
  • Consider presenting more results in a graphical form – tables are clear, yet they do not focus the eye of a potential reader. Additional plots and charts would be highly desirable.
  • The Conclusions section is far too short. Surely the Authors, after performing a series of studies and experiments, have more thoughts and recommendation for other scientists and potential readers.
  • Moreover, I would advise the Authors to include a Future Work chapter, where they would discuss what aspects still need further investigation. This would be an inspiring and valuable source of inspiration for others.
  • The Author Contributions section does not seem properly filled in. Moreover, as far as I am concerned, there were more statements in the template to be filled in and/or edited.

Round 2

Reviewer 2 Report

Thank you for addressing to my suggestions and comments. Currently, the resubmitted version of the manuscript has improved a lot. In my opinion, this paper is ready for publications.
One short remark - consider inserting Fig. 6 in a more sharp form - at present, this graphs has washed out colors.
